# Estimating the burden of severe malarial anaemia and access to hospital care in East Africa

Peter Winskill [1] ✉, Aggrey Dhabangi[2], Titus K. Kwambai[3], Amani Thomas Mori [4,5], Andria Mousa[6] & Lucy C. Okell [1]

Severe malarial anaemia can be fatal if not promptly treated. Hospital studies may under-represent the true burden because cases often occur in settings with poor access to healthcare. We estimate the relationship of community prevalence of malaria infection and severe malarial anaemia with the incidence of severe malarial anaemia cases in hospital, using survey data from 21 countries and hospital data from Kenya, Tanzania and Uganda. The estimated percentage of severe malarial anaemia cases that were hospitalised is low and consistent for Kenya (21% (95% CrI: 7%, 47%)), Tanzania (18% (95% CrI: 5%, 52%)) and Uganda (23% (95% CrI: 9%, 48%)). The majority of severe malarial anaemia cases remain in the community, with the consequent public health burden being contingent upon the severity of these cases. Alongside health system strengthening, research to better understand the spectrum of disease associated with severe malarial anaemia cases in the community is a priority.

Malaria remains one of the most deadly infectious diseases in the world, being responsible for an estimated 619,000 deaths in 2021, mostly in young children[1]. Severe malaria caused by *Plasmodium falciparum* presents as a range of clinical manifestations including, but not limited to, cerebral malaria (CM), respiratory distress syndrome (RDS) and severe malarial anaemia (SMA), all of which can quickly lead to clinical deterioration and death in the absence of prompt treatment[2,3].

SMA is an acute clinical manifestation of severe malaria, defined in children as haemoglobin (Hb) concentration <5 g/dl (or haematocrit <15%) combined with the presence of malaria parasites (with the additional criterion of parasite density threshold of >10,000/μl in the WHO definition although this is not always used in clinical settings)[4]. The pathogenesis of SMA is complex, with the destruction of both non-parasitised and parasitised red blood cells (RBC) and decreased RBC production from the bone marrow contributing to severity[5]. SMA pathogenesis also likely represents a spectrum, being influenced by previous malaria exposure, parasite density, existing anaemia and severe anaemia (both chronic and acute) and its related co-occurrence with other drivers such as helminths, sickle cell and malnutrition[6]. However, the diagnostic definition is broad and differential diagnosis is often difficult or not possible. The progression from uncomplicated malaria to SMA is strongly associated with delayed treatment, highlighting the importance of access to prompt diagnosis and treatment whenever symptoms occur[7]. The burden of SMA is disproportionately concentrated in children under 5 years of age[8], and SMA is associated with high in-hospital mortality rates in excess of 5%[9], with over half of the deaths occurring within the first 24 h of hospital admission[10]. Even short delays to effective treatment can have fatal consequences in the context of severe malaria[7]. In addition, surviving children continue to be at high risk of increased morbidity and mortality even after effective in-hospital treatment of an acute episode[11].

[1]MRC Centre for Global Infectious Disease Analysis, Department of Infectious Disease Epidemiology, Imperial College, London W2 1PG, UK. [2]Child Health and Development Centre, Makerere University College of Health Sciences, Kampala, Uganda. [3]Division of Parasitic Diseases and Malaria, Global Health Center, Centers for Disease Control and Prevention, Kisumu, Kenya. [4]Health Economics Leadership and Translational Ethics Research Group (HELTER), Department of Global Public Health and Primary Care, University of Bergen, Arstadveien 17, 5009 Bergen, Norway. [5]Muhimbili University of Health and Allied Sciences, P.O. Box, 65001 Dar es-Salaam, Tanzania. [6]Department of Infection Biology, London School of Hygiene and Tropical Medicine, London, UK. ✉e-mail: p.winskill@imperial.ac.uk

The burden of SMA in malaria-endemic countries remains high despite efforts to scale up case management and preventive interventions that have led to a decline in malaria and severe malaria (including SMA) over the last 20 years. A recent study using hospital data estimated that there were in excess of 1.5 annual hospital admissions per 1000 children aged 3 months to 9 years old in areas of high malaria prevalence in East Africa[12]. In highly endemic areas, SMA can be present in 20–40% of paediatric hospital admissions and contribute to >50% of in-hospital malaria deaths[10]. In many malaria-endemic settings, it is possible that the observed burden only represents a proportion of the total burden of SMA, with further cases occurring in the community without being reported by hospital surveillance systems[13,14].

Despite its high case fatality rate in children, the relative rarity of SMA at any one time in whole population surveys has made it challenging to assess the true burden of the disease. However, one growing source of additional information about the burden of SMA in the community is large-scale, representative household surveys, which collect the same diagnostic metrics used to define SMA in hospital. An analysis of over 180,000 children aged under 5 years from Demographic Health Surveys (DHS) and Malaria Indicator Surveys (MIS) across 19 countries found that nearly 1% of children testing positive for malaria also had severe anaemia (haemoglobin <5 g/dl)[15], providing clues for burden analysis. Although surveys ask questions about treatment seeking, the limited responses obtained from the small numbers of severe malaria cases make conclusions drawn from these data highly uncertain. Current estimates of the global severe malaria morbidity and mortality have a high degree of uncertainty and rely on case incidence, hospitalisation, and verbal autopsy data alongside expert opinion combined within a framework that includes multiple causes of death[16]. Understanding the true burden of SMA is important to improve the current estimates as well as to inform investments in better access to hospital care and prevention interventions.

In this study, we correlate both household survey and in-hospital data sources using a statistical model of disease incidence and hospital access to estimate the proportion of SMA cases that access hospital care and quantify for the first time the burden of SMA in the community.

## Results

Overall, of the 209,542 children in the household survey data, 18% had malaria, 0.30% had severe anaemia, and 0.15% had both. After adjusting for background levels of severe anaemia in children without malaria in each country, an estimated 0.12% of children had SMA. Our model-fitted observed prevalence of severe malarial anaemia in 0.5- to 5-year-olds ($SMA_{0.5-5}$) in the community, capturing trends with respect to *Plasmodium falciparum* parasite prevalence in 2- to 10-year-olds ($Pf Pr_{2-10}$) as well as between country differences (Fig. 1). Although there was substantial uncertainty in the trends within each country due to low numbers, pooling the data across all countries allowed estimation of a more precise relationship between $Pf Pr_{2-10}$ and SMA. Observed SMA prevalence was estimated to have an increasing relationship with $Pf Pr_{2-10}$, that plateaued or decreased slightly at higher values of $Pf Pr_{2-10}$ (>40%).

After allowing for chance overlap between background severe anaemia and malaria, as well as imperfect hospital access, our model could also capture the observed trends in hospitalised SMA incidence with respect to $Pf Pr_{2-10}$, producing a similar relationship as found by Paton et al.[12] (Fig. 2). We estimate that hospitalised SMA incidence rises with increasing $Pf Pr_{2-10}$, before plateauing or falling slightly at higher values of $Pf Pr_{2-10}$ (>40%). Estimates of annual hospitalised incidence per 1000 children of 0.5, 1, and 1.5 corresponded to median estimates of $Pf Pr_{2-10}$ of 16%, 27% and 40%, respectively. There was substantial uncertainty associated with the estimates at higher levels of $Pf Pr_{2-10}$.

Information in the surveys to inform estimates of the proportion of children with SMA who sought treatment was sparse. In Kenya, Tanzania and Uganda, 0% (0 of 2), 0% (0 of 2) and 40% (2 of 5) were reported to have sought treatment at a government hospital, respectively. Our estimates of the percentage of SMA cases hospitalised were low and consistent for Kenya (21% (95% CrI: 7%, 47%)), Tanzania (18% (95% CrI: 5%, 52%)) and Uganda (23% (95% CrI: 9%, 48%)) (Fig. 3A, Table 1). These estimates imply the presence of a large community burden of SMA approximately 5 times higher than cases observed in hospital (Fig. 3B).

Estimates of total community burden were somewhat sensitive to the definition of SMA used, decreasing when fever (caregiver reported child as having fever in the 2 weeks preceding the survey) was included in the definition, which correspondingly increased the estimated probability of hospitalisation. Results were less sensitive to the malaria diagnostic used and the adjustment for non-malarial severe anaemia. Estimates were most sensitive to the assumed duration of SMA in the community, with a longer duration of SMA decreasing community incidence estimates and therefore increasing the estimate of the proportion of cases hospitalised (Table 1). The median posterior estimate of the percentage of cases hospitalised did not exceed 40% in any of the analyses.

We saw weak evidence for a decrease in the percentage of cases hospitalised with respect to the distance to hospital (Supplementary Fig. S1); however, the range and resolution of distance to hospital data were not enough to identify a strong trend.

We compared our modelled estimates of SMA incidence to those observed during the RTS,S trial as a means of validation. Estimates of the observed SMA incidence in the trial fell between the median estimates of SMA hospitalised incidence and SMA total incidence predicted by our model, overlapping with our uncertainty ranges (Supplementary Fig. S2).

## Discussion

Our analysis relates severe anaemia and malaria measured in representative household surveys with hospitalised SMA incidence and estimates that the percentage of SMA cases that are hospitalised is consistently low at 21% (95% CrI: 7–47%), 18% (5–52%), and 23% (9–48%) for Kenya, Tanzania, and Uganda, respectively (Fig. 3, Table 1). These results indicate that the community burden of SMA may be around 5 times higher than inferred from the hospital data, although considerable uncertainties remain. Our results were robust over a range of sensitivity analyses that explored some key assumptions underlying the work (Table 1).

What does a community SMA case look like? This is a critical question key to understanding the full extent of the community burden, and one that is hard to fully answer given current data. Our estimate of the percentage of cases hospitalised represents a complex set of drivers influencing the likelihood that a child with SMA received hospitalised care. First, it is likely that there is a wide spectrum of syndromes and associated outcomes for children with malaria and severe anaemia. We acknowledge as a limitation that the household surveys we used to estimate community prevalence of SMA did not contain parasite density data, which may have improved the specificity of our case definition. It is likely, for example, that SMA as a result of mild malaria occurring together with chronic anaemia may follow a different course to a more acute anaemia as a result of a malaria infection in a previously healthy child[17,18]. Potentially, not all cases need hospitalisation or transfusion[9], and some will spontaneously recover following treatment from an outpatient clinic or other non-hospital provider[19]. Even when children are seriously ill, SMA appears to be a difficult health condition for caregivers to recognise, which can lead to delays or a complete lack of treatment seeking[20]. This proportion can be significant; for example, in a study in Tanzania, 67% of mothers of symptomatic children with severe anaemia reported that they did not

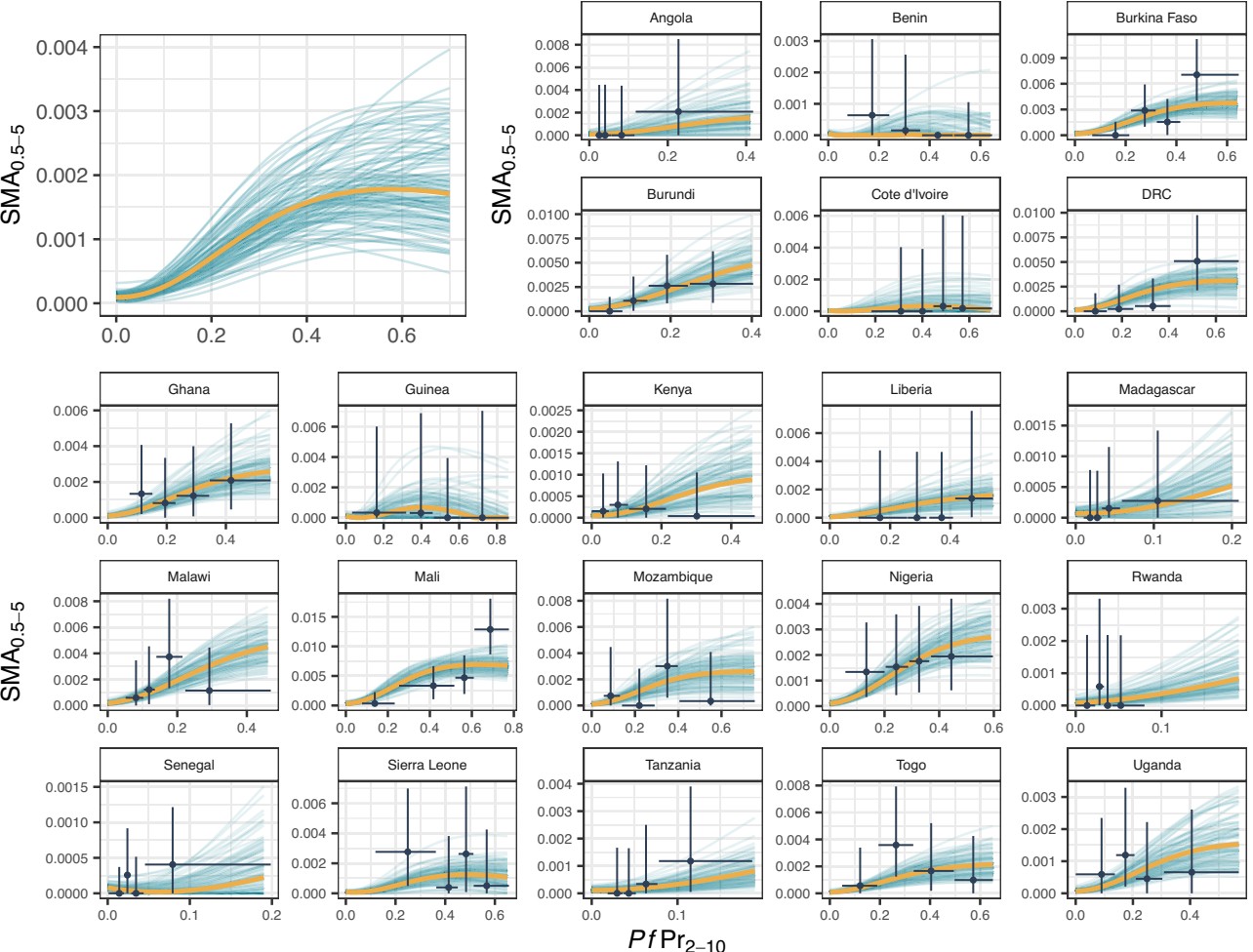

**Fig. 1 | Model fit to data showing the relationship between the prevalence of severe malarial anaemia in 0.5- to 5-year-olds ($SMA_{0.5-5}$) and *Plasmodium falciparum* malaria prevalence in 2- to 10-year-olds ($Pf$Pr$_{2-10}$).** The main plot (top left) shows the overall fitted relationship across all countries (with country-level effects set to zero) and subplots show fits to country data. Orange lines show the posterior median fit (summarised across 120,000 posterior samples), teal lines show 100 draws from the joint posterior and black points, and lines are the DHS data and associated 95% confidence intervals. $SMA_{0.5-5}$ is estimated by adjusting the survey prevalence of malaria and severe anaemia in 0.5- to 5-year-olds by the background prevalence of severe anaemia. Source data are provided as a Source Data file.

think the illness was severe[13]. Even when symptoms are recognised, there remain significant physical and financial barriers to accessing hospital care. Whilst our adjustment for non-malarial severe anaemia is included to account for other drivers of anaemia, such as helminths, sickle cell and malnutrition, significant heterogeneity in the distribution of these co-occurring variables may influence the estimates derived[21]. Malaria is commonly associated with poverty, and in this context, a host of health inequities create barriers that prevent children with SMA from accessing timely and appropriate treatment[13]. Moreover, the referral systems and networks for healthcare seeking for SMA vary widely across countries, for example, being influenced by community health workers or the availability of blood for transfusion. As a result, the generalisability of our findings outside of the settings in the study is uncertain. SMA often occurs alongside other severe malaria syndromes, such as respiratory distress or cerebral malaria, which can increase the probability of hospitalisation and case fatality rate[7,22]. These other types of severe symptoms may be more easily recognised as needing urgent care, although conversely, these cases may also be more likely to die before reaching a hospital. The case fatality of SMA in hospital is in the range of 2–20%[23,24]; this value could be higher in the community due to lack of hospital care or lower if those who do not seek care have less severe disease. As a result, it

remains unclear how the burden of community SMA translates to child health outcomes.

Our findings indicate that a large proportion of SMA is occurring in the community. In line with these findings, and faced with the same data limitations, other studies have also pointed towards significant gaps in hospital access and a large community burden for severe malaria. Camponovo et al. attempted to triangulate World Malaria report estimates of cases, admissions, and deaths at the national level with model predictions to produce estimates of the proportion of cases admitted in hospitals. They estimated that 9% (range: 7–31%), 48% (37–100%), and 63% (51–100%) of patients are admitted in Kenya, Tanzania, and Uganda, respectively, although they stress the need for further studies given that part of their analysis relied on expert opinion[14]. Whilst these ranges overlap with our estimates, the high degree of uncertainty in both makes direct comparisons challenging. The CARAMAL study, which tracked children with suspected severe malaria in the community, found that 58%, 67%, and 48% attended hospital following referral in Uganda, DRC, and Nigeria, respectively, which are higher than our estimates for SMA. However, these were cases that had sought initial treatment and that included all types of severe malaria manifestations in addition to severe anaemia, e.g., cerebral malaria or respiratory distress,

which may be more easily recognised as severe, and also progress more rapidly[25].

More widely, other studies have attempted to harness information from verbal autopsies to address questions about hospital access and the location of death in low-resource settings. These studies provide clues about the community burden of disease. Using verbal autopsy, Fraser et al. estimated that around one-third of 1324 deaths from time-critical conditions screened in South Africa did not seek care before dying[26]. A study by Gill et al. in Lusaka, Zambia, showed that the majority

of respiratory syncytial virus (RSV) deaths in children occurred in the community[27], similarly pointing towards a significant community burden of disease. A study of deaths occurring during pregnancy or childbirth in Burkina Faso and Indonesia showed that 41% (72/174) of deaths occurred at home, with 22% and 8% in each country, respectively, being associated with malaria[28]. Data from the million deaths study in India show that in excess of 80% of deaths in children under 15 years old occur outside of the healthcare facility, with 75% of all deaths occurring at home[29]. Whilst the context differs between these studies, large numbers of deaths occurring at home highlight a significant burden of severe disease in the community, as the results of the current study have also indicated. Overall, our results are consistent with the malaria mortality estimated from verbal autopsies in the community. Efforts to infer total malaria mortality rates[30] in the community and hospital combined have led to estimates that are similar in magnitude to the incidence of hospitalised severe malaria[12]. Since severe malaria case fatality in hospital is in the range of 2–60%, these numbers suggest a significant "hidden" community burden.

Our results are sensitive to the assumed duration of SMA in the community. The relationship between duration and hospitalisation probability is positively correlated (Supplementary Fig. S3). Re-running the analysis with a less informative prior on the duration of SMA resulted in the estimates of the probability of hospitalisation and duration both increasing. However, it is not clear how plausible the posterior estimate of duration (Supplementary Table S2, 74 days (95% CrI: 21, 230 days) versus 41 days (95% CrI: 15, 100) in the main analysis) is in this instance. Although information on the duration of SMA in the community is scant, we do know that SMA cases admitted to hospital usually report to have had symptoms for less than 7 days (e.g., refs. 7,31), and other data from animal models and studies in adult populations indicate similar timescales (Supplementary Information: "Infection timescale"). This may indicate that our original assumption of a short duration is more plausible or could point towards a spectrum of severity associated with those with SMA in the community

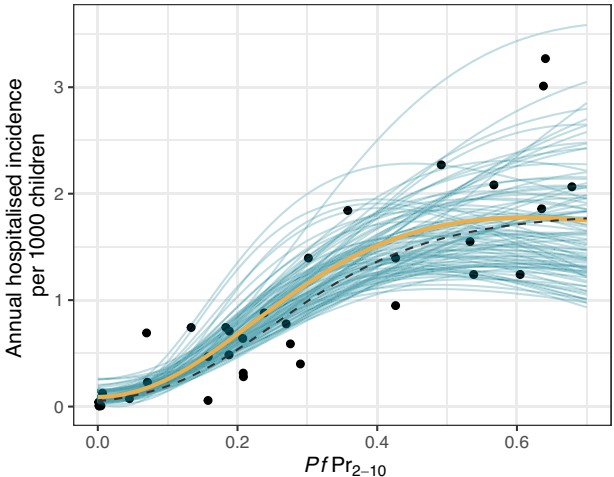

**Fig. 2 | Model fit to data showing the relationship between the prevalence of malaria in 2- to 10-year-olds ($Pf\mathrm{Pr}_{2-10}$) and SMA hospitalisations.** The orange line shows the posterior median fit (summarised across 120,000 posterior samples), teal lines show 100 draws from the joint posterior, black points are the observed hospitalisation data[12], and the black dashed line is the previously fitted model from Paton et al. Source data are provided as a Source Data file.

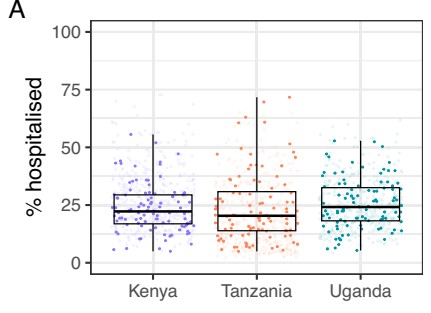

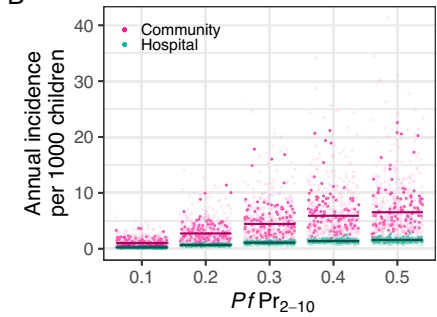

**Fig. 3 | The probability of a child with SMA accessing hospital and estimated community incidence of SMA compared to hospital incidence. A** The estimated posterior distribution of % hospitalised for a child aged 0.25–9 years with SMA in Kenya, Tanzania, and Uganda, box midline indicates the median, shoulders the 25% and 75% percentiles and whiskers the range from the main analysis. Dark points represent samples from the main analysis (100 per country) and faded points

samples across all sensitivity analyses (100 per country). **B** A comparison of the estimated community incidence of SMA with respect to malaria prevalence in 2- to 10-year-olds ($Pf\mathrm{Pr}_{2-10}$) compared to the incidence observed through hospital admissions. Dark points and lines represent 1000 samples and the median estimate, respectively, from the main analysis. Faded points show 1000 samples for each sensitivity analysis. Source data are provided as a Source Data file.

**Table 1 | Summaries of the posterior estimate of the percent of children with SMA hospitalised**

| Country | % Hospitalised (95% CrI) | | | | |
| --- | --- | --- | --- | --- | --- |
| | Main analysis | Sensitivity: fever included in SMA definition | Sensitivity: RDT malaria diagnostic | Sensitivity: no adjustment for non-malarial severe anaemia | Sensitivity: uninformative prior on duration of SMA |
| Kenya | 21 (7, 47) | 29 (10, 60) | 25 (9, 53) | 21 (8, 46) | 30 (10, 60) |
| Tanzania | 18 (5, 52) | 25 (6, 63) | 7 (2, 24) | 15 (4, 39) | 24 (7, 60) |
| Uganda | 23 (9, 48) | 26 (10, 53) | 20 (8, 42) | 21 (8, 42) | 31 (12, 59) |

Median estimates and 95% Credible Intervals (CrI) (in brackets) are shown for each country from the median analysis and sensitivity refits.

where duration might be longer than those with the most acute, severe presentation who may immediately seek hospital care. It is likely that the simple duration estimated is an aggregate across more than one duration distribution representing different clinical manifestations of SMA, however, there are currently neither data available nor resolution in the clinical definition widely used to disentangle any such mixtures if they are present.

Including fever in our definition of survey prevalence reduces our estimate of prevalence and consequently increases our estimates of the percentage of cases hospitalised (Fig. 4). However, this definition may be too restrictive as we know that a proportion of those presenting at hospital with SMA are not febrile[7]. Results were not as sensitive to the malaria diagnostic used or the lack of adjustment for non-malarial severe anaemia (Fig. 4, Supplementary Information: "Sensitivity").

We fitted a hierarchical model to data on the prevalence of malaria and severe anaemia, which combined multiple survey-years for a number of countries. This approach was taken due to the low number of malaria and severe anaemia cases observed in the survey data (306 total across surveys from 21 countries). The upper estimate of the malaria and severe anaemia prevalence with respect to $Pf$ $Pr_{2-10}$ therefore borrows information from data from all 21 countries included, whilst the growth rate and shift parameters were assumed to be the same for all countries. The extent to which these assumptions are true across settings will influence estimates as the trend in a country is not only informed by data from that country but also, to a lesser extent, data from all countries included. We can see this influence in countries included in the hospitalisation fitting where we observe median model estimates at higher $Pf$ $Pr_{2-10}$ exceeds (e.g., Fig. 1. Kenya) or is lower (e.g., Fig. 1. Tanzania) than survey point estimates, albeit with a large degree of uncertainty due to the small amount of data from single countries. The extra complexity in our model compared to that of Paton et al.[12] results in more flexibility in the functional form relating hospitalised incidence to $Pf$ $Pr_{2-10}$. Whilst the resulting fitted functional forms are very similar, the Paton et al. function monotonically increases whilst the fit presented here shows a slight fall above $Pf$ $Pr_{2-10}$ of 40% (Fig. 2) as a result of increasing non-malarial SMA incidental with a malaria infection. This does not strongly impact our estimates of the probability of a child with SMA accessing a hospital but is an area requiring future research to understand more fully. Our analysis may be confounded by the distribution of other factors associated with the prevalence of malaria, anaemia and severe disease that we cannot individually control. These include other infectious diseases (e.g., HIV), parasitic infections (e.g., hookworm), nutritional issues (e.g., Vitamin A or B12 deficiencies), enzymopathies (e.g., G6PD), haemoglobinopathies (e.g., sickle cell anaemia)[5], age (further complicated by difference age-ranges in the population survey and hospitalisation data sets that we adjust for), socio-economic variables and malaria treatment. Furthermore, the hospitalisation data is not representative of the whole country, sampling populations from within 30 km of hospitals and excluding urban areas[12]. Differential case fatality rates in the community and hospital may also introduce bias via death censoring. Given a child has SMA, we assume a constant probability of hospitalisation. More complex models could be proposed, for example, assuming that the probability that SMA leads to severe disease needing hospitalisation is a function of $Pf$ $Pr_{2-10}$. However, we saw a good fit to hospitalisation data using the simple model.

Our findings lead to two key conclusions. First, more work is needed to understand the spectrum of disease associated with SMA cases in the community and how Hb responds throughout the time course of an infection and recovery. This is vital to be able to enumerate the child health impact of community SMA. Second, the results point towards a substantial and largely unobserved burden of SMA occurring in the community and the importance of removing or reducing barriers that erode the probability that severely ill children receive the hospital care that they need.

## Methods
### Inclusion and ethics
Ethical approval for this secondary analysis of data study was granted by the Imperial College Research Ethics Committee (ICREC), ref: 22IC7782.

Our authorship team includes researchers from both malaria-endemic and non-malaria-endemic settings. Together we designed the study to be relevant to the study context and to maximise the contribution and impact of the combined skills, knowledge and experience of all authors.

### Data
We extracted data on malaria status, as determined by microscopy or rapid diagnostic test (RDT), haemoglobin level, reported fever status in the last 2 weeks, and reported hospital (public) attendance from DHS and MIS survey data among children aged 6 to 59 months from 21 countries between 2011 and 2020[32]. Children with haemoglobin levels of <5 g/dl were classified as severely anaemic. The main analysis used microscopy to determine malaria infection as it is indicative of current infection and has a higher parasite density threshold of detection than rapid diagnostic tests (RDTs)[33] and may therefore be a better indicator of severe anaemia caused by malaria. We aimed to relate the prevalence of SMA to the local transmission intensity and therefore estimated the cluster-level prevalence of malaria infection as the average Malaria Atlas Project predicted prevalence among 2- to 10-year-olds ($Pf$PR$_{2-10}$)[34] within a 5 km radius of the cluster geolocations, to account for the random offset of geolocations in the data. Hospitalisation data were extracted from Paton et al.[12] using the measurement-adjusted hospital SMA (defined as Hb <5 g/dl on admission in combination with a positive malaria diagnosis) admission rates (see Paton et al., Supplement Fig. S3A). We

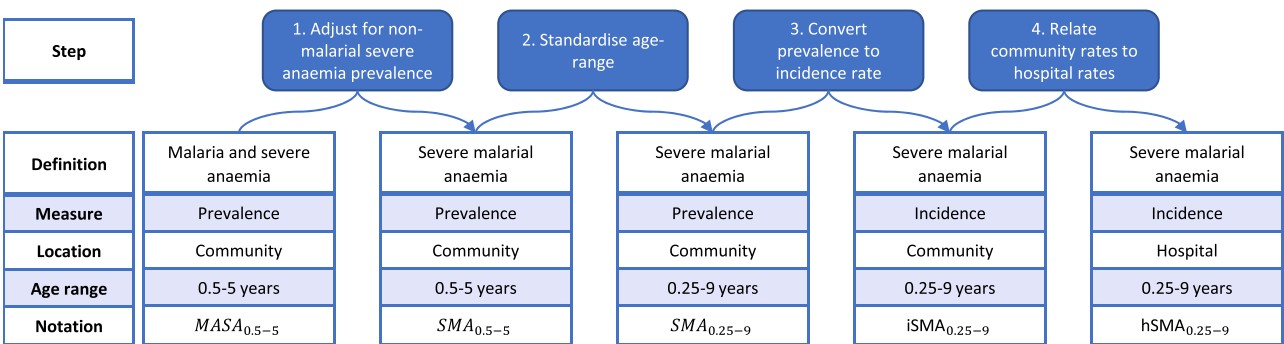

**Fig. 4 | Definitions of elements used in the analysis pathway.** Blue boxes at the top highlight the key adjustments and transformations as each step to move from malaria and severe anaemia prevalence in the survey data to severe malarial anaemia incidence in the hospitalisation data.

also extracted $PfPR_{2-10}$ and distance to the nearest hospital (centre of range, km) estimates for each site.

## The model

We designed a statistical model to correlate the DHS and MIS survey estimates of the prevalence of malaria and severe anaemia in the community with the hospitalisation records capturing the incidence of SMA hospital admissions. Paton et al. found a sigmoidal relationship between community parasite prevalence and incidence of SMA in hospital[12]. We similarly allowed the prevalence of children with both malaria infection and severe anaemia in the community (MASA) to be determined by community parasite prevalence. Additionally, we allowed for background rates of severe anaemia with non-malarial causes, which may co-exist with malaria infection. First, we fitted a hierarchical model to the individual (Hb and malaria status by microscopy) and DHS cluster-level (malaria transmission intensity) data to capture the change in the prevalence of MASA with respect to malaria transmission. We included a country-level random effect to capture systematic differences in this relationship between countries. Following this, we adjusted for a constant country-level background prevalence of non-malarial severe anaemia (Hb <5 g/dl with an accompanying negative malaria test) (Fig. 4, step 1). This accounts for the background rate of severe anaemia, including both acute and chronic causes and adjusts for co-endemic underlying drivers of non-malarial anaemia such as helminths, sickle cell and malnutrition. Next, we age-standardised to match the age range recorded in Paton et al. (3 months–9 years) using a previously defined relationship between age and SMA[12] (Fig. 4, Step 2). We then converted this adjusted prevalence to an incidence rate with a fitted estimate of the duration of an SMA case in the community (Fig. 4, step 3). This incidence was then further adjusted by a distance-dependent hospitalisation probability to fit the hospitalisation data (Fig. 4, step 4). The hospitalisation probability was informed by the proportion of MASA cases reporting to have attended a public hospital in the survey data. We jointly fitted the models of MASA prevalence and hospitalised SMA incidence in a Bayesian framework using the DrJacoby R package[35]. Full mathematical details and descriptions of the parameters fitted are given in the Supplementary Information.

## Sensitivity analyses and validation

We refitted the model four times to determine the influence of four key assumptions on our estimate of the percentage of SMA cases that are hospitalised; (1) severity associated with the definition of SMA in the community, (2) the diagnostic used in the definition of malaria (microscopy or RDT), (3) the adjustment for non-malarial anaemia and (4) our prior specification on the duration of SMA (Supplementary Information: "Sensitivity"). Although few estimates of population incidence of SMA have been published, we were able to compare our model predictions against findings from the RTS,S vaccine trial (Supplementary Information: "Model validation").

## Reporting summary

Further information on research design is available in the Nature Portfolio Reporting Summary linked to this article.

## Data availability

Data used in this analysis are either publicly available in the referenced publication, online (https://malariaatlas.org/) or available online upon registration (https://dhsprogram.com/data/available-datasets.cfm). Source data for figures and tables are provided with this paper. Source data are provided with this paper.

## Code availability

Code for analysis is hosted at https://github.com/mrc-ide/commal (https://doi.org/10.5281/zenodo.8238352).

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

## Acknowledgements

P.W. acknowledges the Imperial College Research Fellowship and the Bill & Melinda Gates Foundation (INV-043624). L.C.O. acknowledges the UK Royal Society Fellowship. The funders of the study had no role in study design, data collection, data analysis, data interpretation, or writing of the report. All authors had full access to all the data in the study and had final responsibility for the decision to submit for publication. The findings and conclusions in this report are those of the authors and do not necessarily represent the official position of the U.S. Centers for Disease Control and Prevention.

## Author contributions

P.W.: Conceptualisation, Methodology, Software, Formal analysis, Writing—Original Draft, Visualisation, Writing—Review & Editing, Data curation, Validation. L.C.O.: Conceptualisation, Methodology, Writing—Review & Editing, Data curation, Validation. A.D., T.K.K., A.T.M., A.M.: Methodology, Writing—Review & Editing.

## Competing interests

The authors declare no competing interests.
