## [Peer Review File · Nature Communications]

REVIEWER COMMENTS

Reviewer #1 (Remarks to the Author):

Okell et al.

Estimating the burden of severe malarial anaemia and access to hospital care in East Africa

REVIEWER COMMENTS

This manuscript presents a modeling study of out-of-hospital incidence of *P. falciparum* severe malarial anemia ("SMA") and health care utilization in children by combining community prevalence data from 21 countries and hospital data from 3 countries, and concludes that up to 4x SMA occurs out of the hospital than in, arguably representing a gap in health care access and case identification. Given the incomplete list of assumptions (e.g., no consideration of chronic vs. acute anemia- their "duration" parameter of anemia does not address this, as these are clinically distinct entities- and meanwhile their model is highly sensitive to the duration parameter), technical concerns about the methods, and failure to explicitly account for coendemic helminths/sickle cell/malnutrition, the authors' claims are not fully substantiated and could misdirect malaria control programs and health policy makers.

Without digging into the supplement its very difficult to follow the Model subsection of the Methods. Equations are not numbered in the supplement. In the supplement, need to use something other than iSMA since "i" is already used for country index.

It is not clear how the authors combine across countries to get the overall relationship between PfPr and MASA (Figure 1) or the overall relationship between hospitalized incidence and PfPr (Figure 2). It seems based on the supplemental information that the estimates for each country ("i") are estimated together to borrow information across countries; however, it's not clear how these individual country estimates (e.g. MASA_0.5-5i) are then aggregated to obtain the overall relationship.

Similarly, it seemed odd that they did quite a rigorous mathematical adjustment for age so that the hospitalization data matched the MASA data, so why not do it for the other data?

The description of the Bernoulli random variables in supplement lines 33, 36, and 41, is confusing. If p_i^{MASA} is a probability itself, then it should not be distributed as Bernoulli- it seems that p_i^{MASA} should take on value 0 or 1 with probability $\text{MASA}_{0.5-5i}$ so the p_i^{MASA} is the random variable denoting whether or not a child in country i has malaria and severe anemia (value 0 or 1) with probability $p_i^{\text{MASA}} = \text{MASA}_{0.5-5i}$?

The authors assume an overall duration of SMA that is drawn from a single distribution- this belies the nature of acute and chronic anemia- the “MASA” -> SMA anemia the investigators are converting in the community -expected to be mostly chronic.

Given the sensitivity of the model outputs to the duration-of-SMA parameter which is what relates the SMA prevalence to incidence, it seems appropriate to have some comparison (at least in the supplement) to some prevalence data of SMA. As of now it seems that in Figure 2 the data points shown are incidence, and the model output is also incidence, and Figure S2 comparing to the RTS,S trial shows the model incidence predictions.

The point of Figure S2 is a bit unclear given that the authors do not describe how the RTS,S data might compare to the hospitalization vs. community level estimates. The model underestimates hospitalization incidence and overestimates community incidence.

Figure 3b which is arguably the “take away” figure from this manuscript is highly problematic and should be removed or drastically redrawn- the authors show no uncertainty around their estimate – important esp since their estimates are sensitive to model parameterization and no doubt varies greatly geographically according to health-seeking behavior, prevalences of other causes of and other factors. The figure is a drastic oversimplification and I doubt its accuracy and generalizability based on the methods used to estimate it and the authors’ neglect of acute vs. chronic anemia, STH, malnutrition, etc. The authors sort of touch on the issue that potentially not every case will require hospitalization and that the progression of the community cases will be possibly varied (lines 139-160) but is just glossed over.

What are the implications if a substantial proportion of the community cases (never hospitalized) of severe malarial anemia were chronic? Hospitalizing children so they can get blood transfusion is not necessary- and can be harmful- in chronic anemia, thus for the sort of SMA they are detecting in the community almost certainly would not benefit from hospitalization and transfusion – see the Cochrane meta-analysis.

Did the authors review helminth data? A major cause of anemia in children in malaria co-endemic areas is soil transmitted helminth (STH) infection- in some series more than half of children – public data on prevalence estimates are available from www.thiswormyworld.org (73 datasets from 165 countries).

Minor comments:

Line 40: I think this definition of SMA is from the 1990s and the parasite threshold for diagnosis has been dropped

Line 45: Consider, “access to prompt diagnosis and treatment”

Line 65 and other parts: Chronic vs. acute anemia in setting of Pf parasitemia warrants more discussion

Line 80: PfPr₂₋₉ is not defined before the first time it is used; MASA_{0.5-5} --the age range is not mentioned in the definition before the first time the abbreviation is used.

Line 187: Case fatality for SM vs. SMA-- cited above for SMA rather than lumping in CM and SMRD (2-60% estimate, vs. 2-20% in the Introduction)

Line 271: Adjusted for country-level non-malarial severe anemia how?

Line 285: Why do the authors think that fever corresponds to severity? This is not in any case definition of severe malaria.

Lacks a schematic that shows generally how the authors progress from the relationship between MASA and PfPr to their “final” estimate of % hospitalization (Figure 3, which has its own problems, above). Table 1 is not explicit.

In the supplement, lines 46-48, the authors mention that convergence was monitored- they should share the results of those convergence metrics in the supplement.

The code for the model and MCMC estimation procedure should be made publicly available on GitHub.

Reviewer #2 (Remarks to the Author):

In this article, the authors have presented an interesting analysis of DHS/MIS surveys to quantify potential severe anemia rates throughout East Africa. While I think this article has significant promise to being a very useful article and is relevant to the Nature Communications audience, and the model is well-described and appropriate, a fair bit of editing/refocusing is needed in the manuscript as-is.

Primarily, the Discussion reads mostly as a related set of points rather than a broader interpretation of the results per se, and needs fairly significant restructuring. I think this is partly because the interpretation of the results is fairly simple as-is. My main suggestion would be to try to identify other results to discuss and contextualize in the Discussion. For example, it seems like the sensitivity analyses would yield new estimates of anemia burden (which is why there's a downstream effect on % of anemia cases that get hospitalized)--it might be worth plotting how these sensitivity analyses change the overall estimate of severe anemia as well.

The discussion also includes a number of paragraphs that don't seem particularly relevant to the results as-is. (such as the two paragraphs beginning "Camponovo et al attempted to triangulate..."). This context may be better in the Intro, but regardless, if this text is going to be kept, the direct importance for interpretation of results of this study needs to be better described.

Also, as far as what the characteristics of community SMA look like (mentioned in the discussion), that actually seems like a potentially interesting analysis to explore with the data. If I understand the modeling process correctly, should be a cluster-level prediction of SMA and prevalence--it may be worth looking into the spatial patterns of both over all 3 countries to look at communities that may have particularly high SMA.

Additional detail of the RTS'S trial would be useful as well. Was the detection of SMA/potential healthcare seeking behavior different from during the DHS/MIS? On a minor related note Figure S2 is a little hard to interpret--do the boxes represent predicted SMA incidence at hospital and hospital + community, and the lines represent potential SMA derived from the RTS'S study using different thresholds? It's unclear which box should be compared with the dotted lines.

Reviewer #1 (Remarks to the Author):

This manuscript presents a modeling study of out-of-hospital incidence of *P. falciparum* severe malarial anemia (“SMA”) and health care utilization in children by combining community prevalence data from 21 countries and hospital data from 3 countries, and concludes that up to 4x SMA occurs out of the hospital than in, arguably representing a gap in health care access and case identification. Given the incomplete list of assumptions (e.g., no consideration of chronic vs. acute anemia- their “duration” parameter of anemia does not address this, as these are clinically distinct entities- and meanwhile their model is highly sensitive to the duration parameter), technical concerns about the methods, and failure to explicitly account for coendemic helminths/sickle cell/malnutrition, the authors’ claims are not fully substantiated and could misdirect malaria control programs and health policy makers.

Regarding the general point of acute and chronic anaemia.

Firstly, a clarification. We agree with the reviewer that the crude prevalence of SMA as indicated by the prevalence of children with malaria and severe anaemia (MASA, Hb<5 & positive malaria test) can be biased by various factors such as severe/chronic anaemia caused by non-malarial factors (e.g. co-endemic helminths/sickle cell/malnutrition) occurring in children who have incidental mild malaria infection. To account for this, in the methods presented we use the observed prevalence of severe anaemia in children who do not have malaria infection to adjust MASA for the background prevalence of severe anaemia (For example, over all survey populations this approach adjusts the MASA estimate of 0.15% down to an SMA estimate of 0.12%). This adjustment accounts for many of the issues raised including:

- 1. The potential for incidental malaria infection in children with Hb<5 where the anaemia has been caused by other factors. This avoids overestimating the measured prevalence of children with severe malarial anaemia (SMA), as defined by test positive and Hb <5.*
- 2. The potential for chronic anaemia, with a long duration, to affect the estimates. With the adjustment, if the population has high rates of non-malarial chronic anaemia, the estimated prevalence of non-malarial severe anaemia will be higher, leading to a stronger downwards adjustment of the estimates of SMA.*
- 3. The potential for specific co-endemic factors such as helminths, sickle cell disease, and malnutrition that cause anaemia to affect the estimates similarly.*

We have included clarifications in the introduction (SMA pathogenesis is also likely represents a spectrum, being influenced by previous malaria exposure, parasite density, existing anaemia and severe anaemia (both chronic and acute) and its related co-occurrence with other drivers such as helminths, sickle cell and malnutrition. However, the diagnostic definition is broad and differential diagnosis is often difficult or not possible.), methods (This accounts for background rate of severe anaemia, including both acute and chronic causes and adjusts for co-endemic underlying drivers of non-malarial anaemia such as helminths, sickle cell and malnutrition) and Figure 4.

Secondly, thank you for the point raised regarding different forms of “true” SMA, for example could there be an acute SMA requiring hospitalisation and a chronic form that does not? The reviewer makes a good point that chronic anaemia, whether caused by mild malaria infection or other non-malarial factors, might develop more gradually and not be caused by acute, severe malaria infection. We are unsure how the reviewer intended the two to be distinguished clinically – perhaps because acute malaria would have higher parasite density and potentially more severe symptoms? We looked for evidence of this distinction in the literature and found that some but not all research studies use a

parasitaemia threshold within the SMA definition. We acknowledge as a limitation that the household surveys used in our analysis did not collect parasite density data which may help to make the case definition more specific (White et al 2018, Watson et al 2022). However, in many clinical settings any parasites + severe anaemia would still meet the definition of SMA and be treated as such. We have added further discussion of this issue in the discussion (We acknowledge as a limitation that the household surveys we used to estimate community prevalence of SMA did not contain parasite density data, which may have improved the specificity of our case definition. It is likely, for example that SMA as a result of mild malaria occurring together with chronic anaemia may follow a different course to a more acute anaemia as a result of a malaria infection in a previously healthy child.). By the strict definition used in this study and widely adopted elsewhere the distinction is not recognised – ie. differential diagnosis is either not possible, or not made. Having reviewed many papers we could not find any evidence that the two are clinically differentiated. We are aware and agree with the reviewer that some children may present with acute-on-chronic anaemia, instead of acute SMA. However, clinically, these two may not be distinct entities e.g a child with IDA (typically a chronic anaemia form) can get malaria, and lapse into SMA.

Estimating a single anaemia duration parameter is a simplification over what may be multiple underlying duration distributions. However, the scope of the data available make disentangling these distributions non-identifiable currently. Our intention was not to imply that children classified as having SMA in the community faced the same clinical course or health outcomes as those hospitalised. To emphasise this and highlight the currently lack of understanding we included a specific paragraph in the discussion of this issue beginning “What does SMA in the community look like?”.

To address these concerns, we have made a number of changes throughout the article:

- We have clarified and highlighted the step adjusting for non-malarial SA both in the methods and in the analysis schematic. We have included more details about what this step may be expected to adjust for and the limitations of this approach (Following this we adjusted for a constant country-level background prevalence of non-malarial severe anaemia (Hb<5 g/dl with an accompanying negative malaria test) (Figure 4, step 1). This accounts for background rate of severe anaemia, including both acute and chronic causes and adjusts for co-endemic underlying drivers of non-malarial anaemia such as helminths, sickle cell and malnutrition, and Figure 4.*
- We have stressed that our single duration parameter may represent a summary over a mixture of (unknown) distributions (It is likely that the simple duration estimated is an aggregate across more than one duration distributions representing different clinical manifestations of SMA, however there are currently neither data available nor resolution in the clinical definition widely used to disentangle any such mixtures if they are present.)*
- We have further added to the “What does SMA in the community look like?” section to stress the uncertainties and gaps in the knowledge here (Lines 158-200).*

Without digging into the supplement its very difficult to follow the Model subsection of the Methods. Equations are not numbered in the supplement. In the supplement, need to use something other than $iSMA$ since "i" is already used for country index.

Thank you for your suggestion. We have now numbered the equations in the SI. We respectfully do not think that changing the name of $iSMA$ is necessary. The "i" at the beginning of $iSMA$ serves an informative purpose by indicating incidence, and we use standard formatting by placing the country index i as a subscript to the lower right of the variable.

It is not clear how the authors combine across countries to get the overall relationship between PfPr and MASA (Figure 1) or the overall relationship between hospitalized incidence and PfPr (Figure 2). It seems based on the supplemental information that the estimates for each country ("i") are estimated together to borrow information across countries; however, it's not clear how these individual country estimates (e.g. MASA_0.5-5i) are then aggregated to obtain the overall relationship.

We have used mixed effects methods to capture the natural variation between countries, whilst estimating "global" values of the parameters across all countries. It is these second-level parameters from which we obtain the overall relationship presented by setting country-level effects to zero. We have now clarified this in the legend text (Figure 1 ... with country-level effects set to zero).

Similarly, it seemed odd that they did quite a rigorous mathematical adjustment for age so that the hospitalization data matched the MASA data, so why not do it for the other data?

We again thank the reviewer for this comment. However, it is not clear to what adjustment or data specifically this comment is directed. The adjustment for age was made due to the availability of consistent data on the distribution with respect to age. If the reviewer would be happy to clarify or provide additional information we would be happy to address more completely.

The description of the Bernoulli random variables in supplement lines 33, 36, and 41, is confusing. If p_i^{MASA} is a probability itself, then it should not be distributed as Bernoulli- it seems that p_i^{MASA} should take on value 0 or 1 with probability MASA_0.5-5i so the p_i^{MASA} is the random variable denoting whether or not a child in country i has malaria and severe anemia (value 0 or 1) with probability $p_i^{\text{MASA}} = \text{MASA}_{0.5-5i}$?

Thank you. We have corrected this error in the SI equations.

The authors assume an overall duration of SMA that is drawn from a single distribution- this belies the nature of acute and chronic anemia- the "MASA" -> SMA anemia the investigators are converting in the community -expected to be mostly chronic.

Please refer to our opening point regarding both the adjustment of MASA for background (including chronic) anaemia and the unidentifiability of further disaggregation of the single duration parameter.

Given the sensitivity of the model outputs to the duration-of-SMA parameter which is what relates the SMA prevalence to incidence, it seems appropriate to have some comparison (at least in the supplement) to some prevalence data of SMA. As of now it seems that in Figure 2 the data points shown are incidence, and the model output is also incidence, and Figure S2 comparing to the RTS,S trial shows the model incidence predictions.

Thank you. Our figure 1 was included as a comparison to the severe malaria and anaemia (MASA) prevalence data. However, we agree with the review in that the SMA prevalence (after adjustment) is a more helpful indicator to present. We have therefore included a version of the figure with SMA as a function of malaria infection prevalence (this is the original plot plus the adjustment for non-malarial severe anaemia).

The point of Figure S2 is a bit unclear given that the authors do not describe how the RTS,S data might compare to the hospitalization vs. community level estimates. The model underestimates hospitalization incidence and overestimates community incidence.

The RTS,S estimates are included as it was the only source of estimates of community SMA incidence that we could find in the published literature. Our modelled estimate of total incidence is higher,

however, the RTS,S observations do fall within the, admittedly wide, uncertainty bounds. We can hypothesise why SMA incidence under ideal trial conditions would be lower than in our estimates based on survey and hospitalisation data (high access to prompt, high quality treatment within the trial setting) but refrain from speculation in our manuscript. We have added further detail to the SI figure legend to clarify (Figure S2).

Figure 3b which is arguably the “take away” figure from this manuscript is highly problematic and should be removed or drastically redrawn- the authors show no uncertainty around their estimate – important esp since their estimates are sensitive to model parameterization and no doubt varies greatly geographically according to health-seeking behavior, prevalences of other causes of and other factors. The figure is a drastic oversimplification and I doubt its accuracy and generalizability based on the methods used to estimate it and the authors’ neglect of acute vs. chronic anemia, STH, malnutrition, etc. The authors sort of touch on the issue that potentially not every case will require hospitalization and that the progression of the community cases will be possibly varied (lines 139-160) but is just glossed over.

This consideration was discussed extensively (lines 139-160) in the original draft. However, we do agree with the reviewers’ material point – that the course and health outcomes of a child with SMA in the community may look different to a child that needs prompt hospitalisation. We have addressed these comments in more detail in the opening statement and in further additions to the discussion. We agree that figure 3b was not a good representation of the findings given the uncertainties associated with the results and have redrawn the figure. (Figure 3)

What are the implications if a substantial proportion of the community cases (never hospitalized) of severe malarial anemia were chronic? Hospitalizing children so they can get blood transfusion is not necessary- and can be harmful- in chronic anemia, thus for the sort of SMA they are detecting in the community almost certainly would not benefit from hospitalization and transfusion – see the Cochrane meta-analysis.

We do not see strong evidence for this claim and believe that differential diagnosis of SMA as chronic vs acute given the current definition would be difficult. It’s possible that a parasite density threshold might help distinguish acute anaemia from chronic but we did not find clear evidence for this. We assume the reviewer means the Cochrane review from 1999 (Meremikwu et al) which found that “[for a] child with severe anaemia who is otherwise stable and not distressed, there is insufficient evidence to know whether the risks of routine blood transfusion outweigh the benefit”. A recent large analysis (Ackerman et al 2020) did not demonstrate any increase in the odds of death as a result of blood transfusion as a function of Hb levels (Figure 3A), in fact transfusion was beneficial even in milder cases of anaemia (up to ~8g/dl) than previously thought. Similarly, Maitland et al found that more rapid transfusion was associated with shorter hospital stay and (non-significant) 46% reduction in mortality in severe anaemia cases (<https://pubmed.ncbi.nlm.nih.gov/31365799/>).

Did the authors review helminth data? A major cause of anemia in children in malaria co-endemic areas is soil transmitted helminth (STH) infection- in some series more than half of children – public data on prevalence estimates are available from www.thiswormyworld.org (73 datasets from 165 countries).

We do not specifically include helminth data in this analysis, or other non-malaria drivers of anaemia. Our adjustment of the background prevalence of severe anaemia is the way that we have approached unbiasing our results with respect to such variables. Please see our opening statement for further details.

Minor comments:

Line 40: I think this definition of SMA is from the 1990s and the parasite threshold for diagnosis has been dropped

We used the WHO reference from 2014 as the standard definition of SMA (Hb <5g/dl and >10,000 parasites / microlitre), but the new WHO Guidelines for malaria 2022 still contain the same density cut off: <https://www.who.int/publications/i/item/guidelines-for-malaria> . We have updated our reference list to use this more recent version. We will appreciate if the reviewer can suggest an alternative official reference with a different definition We agree that in practice in several research studies, this density threshold does not seem to be used (e.g. Kwambai et al 2020 used 5000 parasites/microlitre, Brand et al 2016 used no density cut off <https://journals.plos.org/plosone/article?id=10.1371/journal.pone.0163728>) and now acknowledge this in the introduction when we first define SMA.

Line 45: Consider, “access to prompt diagnosis and treatment”

Suggested change included.

Line 65 and other parts: Chronic vs. acute anemia in setting of Pf parasitemia warrants more discussion

As detailed in our opening point, we have expanded our discussion of this element throughout.

Line 80: PfPr_2-9 is not defined before the first time it is used; MASA_0.5-5 --the age range is not mentioned in the definition before the first time the abbreviation is used.

Definitions have been clarified.

Line 187: Case fatality for SM vs. SMA-- cited above for SMA rather than lumping in CM and SMRD (2-60% estimate, vs. 2-20% in the Introduction)

The discussion point relates to the wider issues of community fatalities for severe malaria as a result of hospital access and we therefore think this distinction is valid.

Line 271: Adjusted for country-level non-malarial severe anemia how?

We have expanded the explanation of this adjustment in the methods text (see also the Supplement for further details) and also made the data processing flow clearer in the associated diagram.

Line 285: Why do the authors think that fever corresponds to severity? This is not in any case definition of severe malaria.

Whilst fever is not in the definition for SMA, fever is known to be linked to the parasite density (for example Plucinski et al 2019). In lieu of parasite density data in the dataset we felt that a sensitivity analysis including fever as a crude proxy for density was worthwhile to explore.

Lacks a schematic that shows generally how the authors progress from the relationship between MASA and PfPr to their “final” estimate of % hospitalization (Figure 3, which has its own problems, above). Table 1 is not explicit.

We have updated the table to include schematic steps shown below in the analysis.

In the supplement, lines 46-48, the authors mention that convergence was monitored- they should share the results of those convergence metrics in the supplement.

Thank you, we have included these in the SI now.

The code for the model and MCMC estimation procedure should be made publicly available on GitHub.

We have included the link to the analysis github repository.

Reviewer #2 (Remarks to the Author):

In this article, the authors have presented an interesting analysis of DHS/MIS surveys to quantify potential severe anemia rates throughout East Africa. While I think this article has significant promise to being a very useful article and is relevant to the Nature Communications audience, and the model is well-described and appropriate, a fair bit of editing/refocusing is needed in the manuscript as-is.

Primarily, the Discussion reads mostly as a related set of points rather than a broader interpretation of the results per se, and needs fairly significant restructuring. I think this is partly because the interpretation of the results is fairly simple as-is. My main suggestion would be to try to identify other results to discuss and contextualize in the Discussion. For example, it seems like the sensitivity analyses would yield new estimates of anemia burden (which is why there's a downstream effect on % of anemia cases that get hospitalized)--it might be worth plotting how these sensitivity analyses change the overall estimate of severe anemia as well.

Thank you for these helpful suggestions. We have made a number of changes throughout the discussion to address this comment. We have added in additional exploration of the specific findings into the discussion. We have also made several changes to Figure 3 to better visualise the uncertainty in both the main analysis as well as the sensitivity analyses.

The discussion also includes a number of paragraphs that don't seem particularly relevant to the results as-is. (such as the two paragraphs beginning "Camponovo et al attempted to triangulate..."). This context may be better in the Intro, but regardless, if this text is going to be kept, the direct importance for interpretation of results of this study needs to be better described.

Thank you for highlighting that elements of the discussion were not well linked to the outcomes of our study. We have made changes throughout the discussion to better integrate our findings with those in the wider literature discussed in this section.

Also, as far as what the characteristics of community SMA look like (mentioned in the discussion), that actually seems like a potentially interesting analysis to explore with the data. If I understand the modeling process correctly, should be a cluster-level prediction of SMA and prevalence--it may be worth looking into the spatial patterns of both over all 3 countries to look at communities that may have particularly high SMA.

We think this is a super idea and believe that one of the key findings of this study is the lack of understanding of community SMA. Given the complexity of the analysis as it stands we do not think that an in depth exploration of these additional factors is within the scope of the current piece of work, but do agree that further analysis in this area is very important.

Additional detail of the RTS'S trial would be useful as well. Was the detection of SMA/potential healthcare seeking behavior different from during the DHS/MIS? On a minor related note Figure S2 is a little hard to interpret--do the boxes represent predicted SMA incidence at hospital and hospital + community, and the lines represent potential SMA derived from the RTS'S study using different thresholds? It's unclear which box should be compared with the dotted lines.

Apologies for the lack of clarity with Figure S2. We have added significant further explanation in the figure legend. We agree that a number of aspects of those data important to the comparison with our results we collected under trial conditions (one could expect for example a higher probability of diagnosis and treatment of any malaria during the trial). However, it is not possible to quantify those differences given the data available and we therefore do not speculate further on specifics.

REVIEWERS' COMMENTS

Reviewer #2 (Remarks to the Author):

I am happy with the responses and edits made by the authors.

Comments from Reviewer #2, on the report submitted by Reviewer #1:

"The remaining disagreements between R1 and the authors seem to do mostly with clinical differences between chronic and acute anemia and whether chronic cases would go to the hospital to begin with. I tend towards agreeing with the authors mentioning that it's difficult to quantitatively assess "chronic" vs "acute" in various populations. I also think as a whole, there is still value towards assessing the overall difference in anemia in the community and in hospital, even though the nature of anemia between the two groups is likely to differ. So, while R1's points are valid and to some degree remain unaddressed, I don't think there is a feasible way to really address them with the data available, and this remains an interesting study despite that. I think it's sufficient that the authors have included R1's views as limitations to the study, and that the study is geared towards modeling anemia as a whole, without distinguishing between acute and chronic per se."

Reviewer's comment(s):

I am happy with the responses and edits made by the authors.

The remaining disagreements between R1 and the authors seem to do mostly with clinical differences between chronic and acute anemia and whether chronic cases would go to the hospital to begin with. I tend towards agreeing with the authors mentioning that it's difficult to quantitatively assess "chronic" vs "acute" in various populations. I also think as a whole, there is still value towards assessing the overall difference in anemia in the community and in hospital, even though the nature of anemia between the two groups is likely to differ. So, while R1's points are valid and to some degree remain unaddressed, I don't think there is a feasible way to really address them with the data available, and this remains an interesting study despite that. I think it's sufficient that the authors have included R1's views as limitations to the study, and that the study is geared towards modeling anemia as a whole, without distinguishing between acute and chronic per se.

Response:

Thank you for the follow up to our initial responses. We have made some further adjustments to ensure that R1's comments are acknowledged in the limitation.